# *Kmt2c*/*Mll3* Haploinsufficiency Causes Autism-like Behavioral Deficits in Mice

**DOI:** 10.3390/biom15111547

**Published:** 2025-11-04

**Authors:** Kaijie Ma, Maria Webb, Haniya Hayder, Luye Qin

**Affiliations:** 1Division of Biomedical and Translational Sciences, Sanford School of Medicine, University of South Dakota, Vermillion, SD 57069, USA; kaijie.ma@usd.edu; 2School of Health Sciences, University of South Dakota, Vermillion, SD 57069, USA; maria.webb@coyotes.usd.edu; 3Department of Biomedical Engineering, College of Arts and Sciences, University of South Dakota, Vermillion, SD 57069, USA; haniya.hayder@coyotes.usd.edu

**Keywords:** autism spectrum disorder, *KMT2C*, social behaviors, sex, cognition

## Abstract

*KMT2C* (histone lysine N-methyltransferase 2C, also known as *MML3*, myeloid/lymphoid or mixed-lineage leukemia 3) is a causal gene for Kleefstra syndrome 2, a rare neurodevelopmental disorder. Recent human genetic studies have identified it as a high-risk gene for autism spectrum disorder (ASD), with 79% of patients harboring *KMT2C* variants having ASD. However, the causal link between *KMT2C* haploinsufficiency and ASD remains unclear. *KMT2C*/*MLL3* encodes a histone methyltransferase, a core protein of the KMT2C/D COMPASS (complex proteins associated with Set1) complex, which plays fundamental roles in chromatin modification, occupancy, and gene expression. The expression of *KMT2C*/*Kmt2c* peaks during the developmental period in the human/mouse brain, which indicates the critical roles of *KMT2C*/*Kmt2c* in neurodevelopment. Here, we investigated the impact of germline *Kmt2c* haploinsufficiency on autism-like behavioral deficits in mice, which modeled humans carrying diverse *KMT2C* variants. Compared with Kmt2c^+/+^ controls, *Kmt2c* haploinsufficiency mice had normal motor function without anxiety-like behaviors. Notably, *Kmt2c* haploinsufficiency mice exhibited autism-like social deficits and increased self-grooming in both males and females, which recapitulated the core phenotypes of ASD patients. Novel object recognition and spatial memory deficits were observed in male and female *Kmt2c* haploinsufficiency mice. This study reveals a causal link between *Kmt2c* haploinsufficiency and ASD-like behavioral deficits. These germline *Kmt2c* haploinsufficiency mice can be used for further studying the molecular mechanisms and developing therapeutic interventions for *KMT2C* haploinsufficiency-associated behavioral deficits.

## 1. Introduction

Autism spectrum disorder (ASD) is a complex developmental disorder with unclear etiology. Growing evidence suggests that the interaction between genetic and environmental factors contributes to the occurrence of ASD [1,2]. Meta-analysis from human studies shows that genetic factors play major roles in ASD, given that ASD has a higher concordance in twins [3,4].

Human genetic studies have identified risk genes for ASD, which are enriched in gene expression or neuronal communication (Simons Foundation Autism Research Initiative, SFARI) [5,6,7]. *KMT2C* (histone lysine N-methyltransferase 2C, also known as *MML3*, myeloid/lymphoid or mixed-lineage leukemia 3) located on human chromosomal band 7q36.1, is a causal gene for Kleefstra syndrome 2 (a rare neurodevelopmental disorder) and a risk gene for ASD. Humans carrying *KMT2C* variants have a spectrum of clinical manifestations named *KMT2C*-related syndrome, including neurodevelopmental delay (NDD), intellectual disability (ID), schizophrenia, seizures, and ASD [5,6,8,9,10,11,12]. Despite the discovery of *KMT2C* as a top-ranking ASD risk gene, the causal link between *KMT2C* haploinsufficiency and ASD remains unclear, and little is known about the molecular mechanisms underlying the behavioral deficits.

*KMT2C* encodes a histone methyltransferase, a core protein of the KMT2C/D COMPASS (complex proteins associated with Set1) complex, which plays fundamental roles in chromatin modification, occupancy, and gene expression via catalyzing histone 3 lysine (K) 4 (H3K4) methylation in the promoters and enhancers [13,14,15]. *KMT2C* is also a member of the ASC-2/NCOA6 complex (ASCOM), which involves transcriptional coactivation [16]. As a tumor suppressor, *KMT2C*/*MLL3* is linked to various cancers, including leukemia, breast, and bladder cancer [17,18,19]. Recent studies showed that mice with brain-specific knockout of *Kmt2c* or a heterozygous frameshift mutation of *Kmt2c* displayed autism-like behaviors [20,21]. Here, we characterized the impact of *Kmt2c* haploinsufficiency on autism-like behavioral deficits by using a different global *Kmt2c* haploinsufficiency mouse model, which modeled individuals with *KMT2C* haploinsufficiency caused by inherited, *de novo* mutations, or deletions. The latest data from the CDC (Centers for Disease Control and Prevention) show that ASD affects 1 in 32 children in the US. Males are more vulnerable to having ASD than females. Therefore, sex as a biological variable was incorporated in this study [22].

## 2. Materials and Methods

### 2.1. Protein–Protein Interaction (PPI) Network and Functional Annotation Analysis

The PPI network associated with *KMT2C* was constructed using the STRING database (version 12.0). The first shell interactors in the PPI network were used for biological process (gene ontology) enrichment analysis.

### 2.2. Animal Care and Husbandry

The use of animals and procedures performed was approved by the Institutional Animal Care and Use Committee of Sanford School of Medicine, University of South Dakota. *Kmt2c* haploinsufficiency (Kmt2c^+/−^) mice were a kind of gift from Dr. Jeffrey Magee (Washington University School of Medicine at St. Louis) [14]. A germline premature stop codon was produced in exon 14 due to a 5-base pair deletion (CATGG). Animals were group-housed (n = 4–5) in standard cages and were kept on a 12 h light-dark cycle in a temperature-controlled room. Food and water were available ad libitum. Male and female Kmt2c^+/−^ mice and sex- and age-matched Kmt2c^+/+^ littermates were used (6–7 weeks old), which were derived from Kmt2c^+/−^ breeding pairs. All behavioral assays were carried out by investigators in a blind fashion (with no prior knowledge of genotypes).

### 2.3. Behavioral Tests

Hindlimb clasping test is a behavioral assessment to evaluate motor function. In brief, mice were gently lifted by the tails, and the hindlimb posture was observed for 30 s. A scoring scale was used to quantify the degree of clasping, as described [23]. Other behavioral tests were performed as in our previous studies [24,25,26]. See Appendix A for details regarding other behavioral tests.

### 2.4. Statistical Analysis

The sample size of each group was calculated to detect behavioral differences based on predicting detectable differences to reach a power of 0.80 at a significance level of 0.05 by power analyses in G*Power software 3.1.9.7. GraphPad software Prism 10.0 (GraphPad Software, La Jolla, CA, USA) was used for statistical comparisons. Differences between more than two groups were assessed with two-way or three-way ANOVA, followed by post hoc Bonferroni tests for multiple comparisons. Data were presented as the mean ± SEM.

## 3. Results

### 3.1. KMT2C/Kmt2c Is Spatiotemporally Expressed in the Human/Mouse Brain

To determine the expression of *KMT2C* in the human brain, we searched it in the Human Brain Transcriptome (HBT) dataset (https://hbatlas.org, accessed on 20 September 2025) [27,28]. As shown in Figure 1A, *KMT2C* is spatiotemporally expressed in the brain, including the neocortex (NCX), hippocampus (HIP), striatum (STR), amygdala (AMY), mediodorsal nucleus of the thalamus (MD), and cerebellar cortex (CBC). The peak level of *KMT2C* is found during the early fetal to late mid-fetal stage in fetal development. To determine the expression of *Kmt2c* in the mouse brain, we searched it in the Allen Developing Mouse Brain Atlas (https://developingmouse.brain-map.org, accessed on 20 September 2025). *Kmt2c* is spatiotemporally expressed in the mouse brain. The peak level of *Kmt2c* is found in P4 mice (Figure 1B). These data indicate the critical roles of *KMT2C*/*Kmt2c* in neurodevelopment.

As shown in Figure 2, human *KMT2C*/*MLL3* encodes a protein of 4911 amino acids with several key domains, including AT-hook, PHD (plant homologous), SET, and post-SET domains, which are involved in binding to the minor groove of DNA, recognition of unmodified or methylated lysine in histone 3, and methylation of histones on lysine to modify chromatin for transcription [29]. In the SFARI database, there are 10 types of *KMT2C* variants reported, including copy number gain, copy number loss, frameshift variant, intron variant, missense variant, splice region variant, splice site variant, stop gained variant, synonymous variant, and translocation. Among them, the majority are frameshift variants (41), missense variants (59), and stop gained variants (32) in the coding region.

### 3.2. The Biological Roles of KMT2C in the Brain

To determine the biological roles of *KMT2C* and its mechanistic link to brain disorders, we did a PPI network analysis using the STRING database. The minimum required interaction score was set at highest confidence (0.9). As shown in Figure 3A, the PPI network identified the maximal first shell interactors with *KMT2C*. The number of nodes was 23, and the number of edges (both functional and physical protein associations) was 163 (PPI enrichment *p*-value: <1.0 × 10^−16^). The top proteins with higher scores included RBBP5, ASH2L, DPY30, and NCOA6, which are the members of the KMT2C/D COMPASS or ASC-2/NCOA6 complex. The second shell interactors with *KMT2C* were not shown. Biological process (gene ontology) analysis revealed that the first shell interactors were mostly enriched in H3-K4 methylation, chromatin assembly, nucleosome assembly, and chromatin remodeling (Figure 3B). These results suggest a multifaceted role for *KMT2C* in the regulation of transcription.

### 3.3. Kmt2c Haploinsufficiency Mice Have Normal Motor Function Without Anxiety-like Behaviors

87% of individuals carrying *KMT2C* variants displayed gross motor delay; however, all individuals older than 3.5 years of age achieved independent walking [12]. The impact of *Kmt2c* haploinsufficiency on motor function in mice was examined at 6–7 weeks old, which is equivalent to a human in teenage years [30]. As shown in Figure 4A, in a hindlimb clasping test, male and female Kmt2c^+/+^ mice showed normal extension reflexes in the hindlimbs. Hindlimb clasping was not observed in male and female Kmt2c^+/−^ mice, indicating normal motor function. Then, we examined the basic locomotion function, the ability of mice to freely move. Male and female Kmt2c^+/−^ mice showed similar distance traveled (*F*
_Genotype (1, 46)_ = 0.003, *p* = 0.95; *F*
_Sex (1, 46)_ = 0.0004, *p* = 0.98) and speed (*F*
_Genotype (1, 46)_ = 0.003, *p* = 0.95; *F*
_Sex (1, 46)_ = 0.0004, *p* = 0.98) with male and female Kmt2c^+/+^ mice in the open field test (Figure 4B,C). To further examine whether *Kmt2c* haploinsufficiency affects motor coordination, we performed a rotarod test. Male and female Kmt2c^+/-^ mice had similar latency to fall (*F*
_Genotype (1, 46)_ = 0.11, *p* = 0.74; *F*
_Sex (1, 46)_ = 0.0007, *p* = 0.98) (Figure 4D), demonstrating that *Kmt2c* haploinsufficiency does not affect motor function.

To determine the impact of *Kmt2c* haploinsufficiency on anxiety, a comorbidity of ASD, we did an open-field test. Kmt2c^+/−^ mice displayed a similar time (*F*
_Genotype (1, 46)_ = 0.0005, *p* = 0.98; *F*
_Sex (1, 46)_ = 0.22, *p* = 0.64) spent and number of entries (*F*
_Genotype (1, 46)_ = 0.37, *p* = 0.54; *F*
_Sex (1, 46)_ = 0.15, *p* = 0.70) in the center (Figure 5A,B), compared to Kmt2c^+/+^ mice. The similar anxiety index in Kmt2c^+/+^ and Kmt2c^+/−^ mice (*F*
_Genotype (1, 46)_ = 0.0005, *p* = 0.98; *F*
_Sex (1, 46)_ = 0.22, *p* = 0.64) (Figure 5C,D) indicates that *Kmt2c* haploinsufficiency does not induce anxiety-like behaviors under a mild stress condition. To further examine if *Kmt2c* haploinsufficiency induces anxiety-like behaviors under a relatively high stress condition, we performed an elevated plus maze (EPM) test [31]. Kmt2c^+/−^ mice spent a similar time (*F* _Genotype (1, 46)_ = 0.05, *p* = 0.82, *F* _Sex (1, 46)_ = 0.02, *p* = 0.89) and number of entries (*F*
_Genotype (1, 46)_ = 0.08, *p* = 0.78, *F*
_Sex (1, 46)_ = 1.17, *p* = 0.29) in the open arms with Kmt2c^+/+^ mice (Figure 5E,F). Kmt2c^+/+^ and Kmt2c^+/−^ mice had similar anxiety index in the EPM test (*F*
_Genotype (1, 46)_ = 0.008, *p* = 0.93; *F*
_Sex (1, 46)_ = 0.04, *p* = 0.84) (Figure 5G,H), which suggests that *Kmt2c* haploinsufficiency has no effects on anxiety-like behaviors under a relatively high stress condition.

### 3.4. Kmt2c Haploinsufficiency Mice Exhibit Autism-like Behavioral Deficits

79% of patients with *KMT2C*-related syndrome had autism [12]. To determine whether *Kmt2c* haploinsufficiency causes autism-like social deficits, male and female Kmt2c^+/−^ and Kmt2c^+/+^ mice were subjected to the three-chamber social interaction assay [24,32]. As shown in Figure 6A, Kmt2c^+/+^ mice spent significantly more time exploring the social stimulus over the non-social object, while Kmt2c^+/−^ displayed reduced preference for the social stimulus (Male Kmt2c^+/+^: social: 133.3 ± 8.8 s, nonsocial: 43.4 ± 3.0 s, n = 12; Female Kmt2c^+/+^: social: 132.2 ± 7.9 s, nonsocial: 44.7 ± 3.7 s, n = 12; Male Kmt2c^+/−^: social: 97.1 ± 4.9 s, nonsocial: 68.7 ± 4.1 s, n = 13; Female Kmt2c^+/−^: social: 95.2 ± 6.2 s, nonsocial: 67 ± 4.6 s, n = 13, *F*
_Soc vs. NS × Genotype (1, 92)_ = 56.1, *p* < 0.0001). Consistently, male and female Kmt2c^+/−^ exhibited a significantly reduced social preference index compared to Kmt2c^+/+^ mice (Male Kmt2c^+/+^: 49.5% ± 4.2%, n = 12; Female Kmt2c^+/+^: 49.2% ± 3.6%, n = 12; Male Kmt2c^+/−^: 17.3% ± 3.2%, n = 13; Female Kmt2c^+/−^: 17.2% ± 2.7%, n = 13. *F*
_Genotype (1, 46)_ = 87.5, *p* < 0.0001, *F*
_Sex (1, 46)_ = 0.005, *p* = 0.95) (Figure 6B,C). The lower time exploring the social stimulus and social preference index in Kmt2c^+/−^ mice suggests that *Kmt2c* haploinsufficiency causes autism-like social deficits in male and female mice.

Self-grooming is an innate behavior in rodents and is used as an indication of compulsive and repetitive behavior [33]. Male and female Kmt2c^+/−^ mice spent significantly more time engaged in self-grooming compared to male and female Kmt2c^+/+^ mice (Male Kmt2c^+/+^: 24.6 ± 3.0 s, n = 12; Female Kmt2c^+/+^: 26.9 ± 3.1 s, n = 12; male Kmt2c^+/−^: 48.2 ± 5.8 s, n = 13, Female Kmt2c^+/−^: 52.9 ± 6.1 s, n = 13. *F*
_Genotype (1, 46)_ = 26.2, *p* < 0.0001, *F*
_Sex (1, 46)_ = 0.5, *p* = 0.47) (Figure 6D). These results demonstrated that *Kmt2c* haploinsufficiency induces autism-like repetitive & restrictive behaviors in males and females.

### 3.5. Kmt2c Haploinsufficiency Mice Display Cognitive Deficits

86% of patients with *KMT2C*-related syndrome had intellectual disability with varied severity [12]. To assess the impact of *Kmt2c* haploinsufficiency on cognition, we first did a novel object recognition (NOR) test to examine whether *Kmt2c* haploinsufficiency could impair novel object recognition memory. 5 min after initial familiarization with two identical objects in the habituated arena, the mice were allowed to explore the same arena in the presence of a familiar object and a novel object. As shown in Figure 7A, male and female Kmt2c^+/+^ mice spent significantly more time exploring the novel object over the familiar object, while male and female Kmt2c^+/−^ mice lacked a preference for the novel object (Male Kmt2c^+/+^: novel object: 34.9 ± 7.2 s, familiar object: 10.8 ± 1.1 s, n = 12; Female Kmt2c^+/+^: novel object: 33.4 ± 4.2 s, familiar object: 13.3 ± 2.8 s, n = 12; Male Kmt2c^+/−^: novel object: 21.9 ± 3.4 s, familiar object: 14.9 ± 1.6 s, n = 13; Female Kmt2c^+/−^: novel object: 22.8 ± 2.9 s, familiar object: 15.6 ± 1.4 s, n = 13, *F*
_Object × Genotype (1, 92)_ = 9.0, *p* = 0.0035). Male and female Kmt2c^+/−^ mice showed similar entry times to the novel object or the familiar object and distance traveled with male and female Kmt2c^+/+^ mice (Figure 7B,C). The lower discrimination index in male and female Kmt2c^+/−^ mice indicates *Kmt2c* haploinsufficiency impairs novel object recognition memory (Male Kmt2c^+/+^: 0.44 ± 0.06, n = 12; Female Kmt2c^+/+^: 0.44 ± 0.06, n = 12; Male Kmt2c^+/−^: 0.16 ± 0.05, n = 13; Female Kmt2c^+/−^: 0.15 ± 0.05, n = 13. *F* _Genotype (1, 46)_ = 26.5, *p* < 0.001; *F* _Sex (1, 46)_ = 0.02, *p* = 0.89) (Figure 7D,E).

To examine whether *Kmt2c* haploinsufficiency affects spatial memory, we performed a Barnes maze test [24,34]. 15 min after two learning phases finding the correct hole and entering the escape box (information acquisition), the mice were allowed to explore the same platform in the absence of the escape box under the correct hole in the memory phase (information retention and retrieval). There were no differences in total investigation time (T1 + T2), distance traveled, and entry times to the correct hole between Kmt2c^+/+^ and Kmt2c^+/−^ mice (Figure 8A–C). Kmt2c^+/−^ mice displayed significantly lower spatial memory index (T1/T2) (*F* _Genotype (1, 46)_ = 32.3, *p* < 0.001; *F* _Sex (1, 46)_ = 0.02, *p* = 0.89), compared with Kmt2c^+/+^ mice (Figure 8D,E), which demonstrates that *Kmt2c* haploinsufficiency impairs spatial memory.

## 4. Discussion

In this study, we characterized the autism-like behavioral deficits in a germline *Kmt2c* haploinsufficiency mouse model. *Kmt2c* haploinsufficiency mice exhibited autism-like social deficits and increased self-grooming, which recapitulated the core phenotypes of ASD patients carrying *KMT2C* variants and confirmed the causal link between *KMT2C* haploinsufficiency and ASD. Furthermore, we identified that *Kmt2c* haploinsufficiency caused cognitive impairments in mice, confirming that ID is a key symptom of Kleefstra syndrome 2 and a major comorbidity of ASD [35].

The P4 mice are still in the developmental stage, which corresponds to the late stage of the second trimester in human gestation, at around the 21st–24th week of gestation [30,36]. Consistently, transcriptomic studies from humans and mice showed the peak expression of *KMT2C*/*Kmt2c* in the brain during the similar developmental time window. The perinatal lethality of Kmt2c^-/-^ mice further confirmed the essential role of *KMT2C* in brain development [14,37,38]. H3K4 methylation is related to transcriptional activation via three types of modifications: mono-, di-, and trimethylation (H3K4me1, H3K4me2, and H3K4me3). Dysregulation of H3K4 methylation associated with genetic risks is implicated in neurodevelopmental disorders, including ASD [39,40]. *KMT2C* is well known for transcriptional regulation via catalyzing H3K4me1 and H3K4me2 [38]. Recent chromatin immunoprecipitation followed by sequencing (ChIP-seq) analysis showed that *KMT2C* peaks colocalized with H3K4me3 peaks [21], which is consistent with the gene ontology analysis of the first shell interactors of *KMT2C*. Neuronal-specific H3K4me3 peaks in human PFC are enriched in synaptic function and conserved in chimpanzee, macaque, and mouse [41,42], suggesting the key role of *KMT2C* in transcriptional regulation of genes involved in neuronal communications.

The phenotypes caused by *KMT2C* mutations are extensive heterogeneity [12]. Male and female Kmt2c^+/−^ mice had normal motor function without anxiety-like behaviors. ASD is characterized by persistent deficits in social communication & interaction, and restricted & repetitive patterns of behavior, interests, or activities. Male and female Kmt2c^+/−^ mice displayed significantly decreased time spent with social stimulus, diminished social preference index, and increased self-grooming, which are consistent with the reports from other groups by using different *Kmt2c* deficiency mouse models [20,21]. Even though the prevalence of ASD is higher in males than in females [43,44], the similar social behavioral deficits and self-grooming in male and female Kmt2c^+/−^ mice indicate *KMT2C* haploinsufficiency affects both males and females. ASD patients have impaired communication as early as age 1–2, preceding clinical diagnosis at about 4–5 years old. The limitation of this study is the focus on one time window but not developmental trajectories. Further longitudinal characterization of behaviors would help identify the impact of *Kmt2c* haploinsufficiency on developmental milestones.

ID is one of the key features of *KMT2C*-related syndrome [12], and 70% of ASD patients have ID [35]. Trithorax-related (trr) is the *KMT2C* ortholog in *Drosophila* (fruit fly). Knockdown of *trr* in the mushroom body (MB), the learning and memory center of the fly brain, impaired short-term memory, indicating *KMT2C*/*MLL3* are required for short-term memory. The lack of gross morphological defects in the MB upon *trr* knockdown suggested that trr-mediated transcriptional activation related to MB neuronal functions was diminished [8]. Kmt2c^+/−^ mice displayed decreased discrimination index in the novel object recognition test and spatial memory index in the Barnes maze test, which indicates that *Kmt2c* haploinsufficiency affects the brain regions related to object recognition memory formation, such as the perirhinal cortex [45], and spatial memory, such as the hippocampus and PFC. Lysine-specific histone demethylase 1 (LSD1) inhibitors are developed for cancer therapies. A recent study showed that vafidemstat, an LSD1 inhibitor, could ameliorate impairments in sociality but not working memory in a Kmt2c^+/fs^ mouse model [21]. However, transitional antineoplastic drugs to children with ASD may cause untoward effects.

The prefrontal cortex (PFC) is a hub brain region for “high level” executive functions [46,47] such as social behaviors, emotion, and cognition, which is impaired in ASD patients and mouse models of autism [25,48,49]. Hyperactivity of the PFC has been thought of as a key pathogenesis of social deficits in ASD [50,51,52,53]. In a rat cortical neuronal culture study, knockdown of *Kmt2c* increased neuronal excitability via altered intrinsic property, excitatory and inhibitory synaptic inputs. RNA sequencing of *Kmt2c*-deficient neuronal networks at DIV 20 (days in vitro) showed that the differentially expressed genes (most of them were downregulated genes) were enriched in ion transmembrane transport and chemical synaptic transmission [54]. However, Nakamura et al. performed RNA-seq of mice bulk adult forebrain, mainly including the prefrontal cortex, and found that *Kmt2c* haploinsufficiency upregulated genes were enriched in synapse, ion transport, cell projection and morphogenesis, while *Kmt2c* haploinsufficiency downregulated genes were associated with ribosome and cell death [21]. PFC sends out glutamatergic and GABAergic transmission to downstream targets [55,56] and receives ascending synaptic inputs from multiple brain regions via bottom–up innervation [57,58,59,60]. The striatum is a brain region important for stereotypic behaviors, which was found to be defective in *Shank3*-deficient autism mouse models [61,62,63]. The neural mechanisms that drive social deficits and increased self-grooming in Kmt2c^+/−^ mice at cellular and circuit levels need to be further investigated.

## 5. Conclusions

In summary, *Kmt2c* haploinsufficiency resulted in autistic-like behaviors in both males and females. The face validity of these mice builds a strong basis for deep mechanistic studies and developing novel, safe therapeutic interventions for ASD patients with *KMT2C* variants.

## Figures and Tables

**Figure 1 biomolecules-15-01547-f001:**
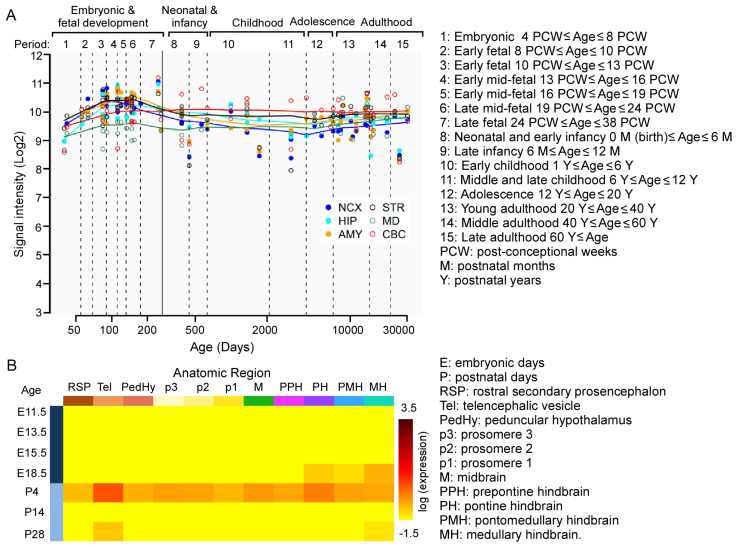
The spatiotemporal expression of *KMT2C*/*Kmt2c* in the human/mouse brain. (**A**) The spatiotemporal expression of *KMT2C* in the human brain. Reprinted and modified from the Human Brain Transcriptome dataset (https://hbatlas.org, accessed on 20 September 2025). (**B**) The spatiotemporal expression of *Kmt2c* in the mouse brain. Reprinted from the Allen Developing Mouse Brain Atlas (https://developingmouse.brain-map.org, accessed on 20 September 2025).

**Figure 2 biomolecules-15-01547-f002:**
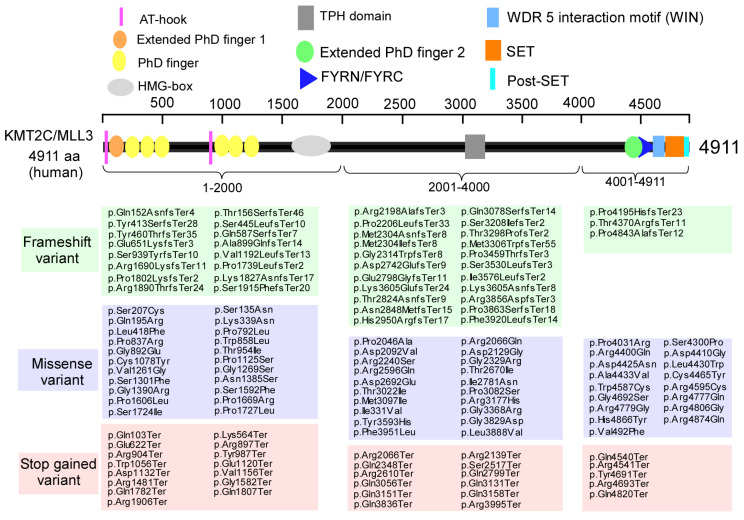
*KMT2C* variants are identified in humans. Domain organization of human *KMT2C*. AT-hook: adenosine-thymidine-hook; PHD: plant homeodomain; HMG: high-mobility group: TPH: Trichohyalin-plectin-homology; FYRN/FYRC, phenylalanine and tyrosine rich region (N- and C-terminal); SET, Su(var)3–9, Enhancer-of-zeste and Trithorax; Post-SET, C-terminal of SET. Modified from the SFARI dataset (https://gene.sfari.org, accessed on 20 September 2025).

**Figure 3 biomolecules-15-01547-f003:**
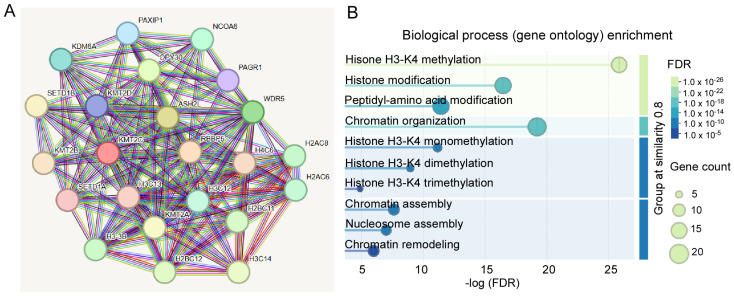
PPI network analysis of *KMT2C*. (**A**) The PPI network analysis showing the first shell interactors with *KMT2C*. (**B**) Biological process (gene ontology) enrichment analysis of first shell interactors. Reprinted and modified from the STRING database (https://string-db.org, accessed on 20 September 2025).

**Figure 4 biomolecules-15-01547-f004:**
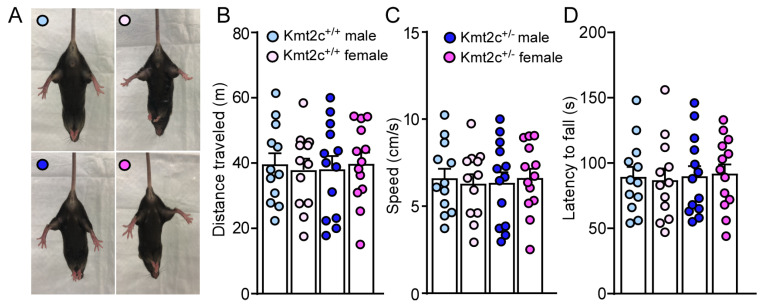
*Kmt2c* haploinsufficiency mice have normal motor function. (**A**) Representative photos showing the hindlimb postures of male and female Kmt2c^+/+^ and Kmt2c^+/−^ mice in the hindlimb clasping test. Bar graphs showing total distance traveled (**B**), speed (**C**) during the open field test and latency to fall (**D**) during the rotarod test of male and female Kmt2c^+/+^ and Kmt2c^+/−^ mice. (**B**–**D**): two-way ANOVA. Male Kmt2c^+/+^ mice: n = 12; Female Kmt2c^+/+^ mice: n = 12; Male Kmt2c^+/−^ mice: n = 13; Female Kmt2c^+/−^ mice: n = 13.

**Figure 5 biomolecules-15-01547-f005:**
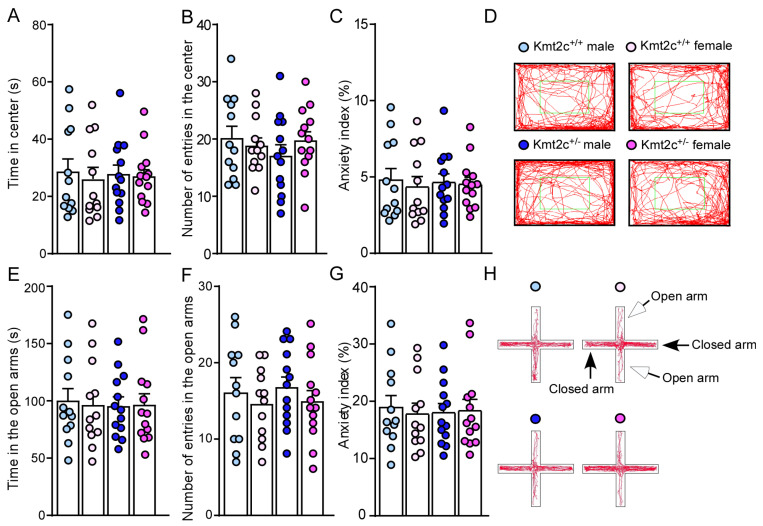
*Kmt2c* haploinsufficiency mice do not have anxiety-like behaviors. Bar graphs showing time spent (**A**), number of entries (**B**) in the center, (**C**) anxiety index, and representative trajectory diagrams (**D**) of Kmt2c^+/+^ and Kmt2c^+/−^ mice in the open field test. The anxiety index in the open field test was calculated as: (time spent in the center)/(total time) × 100%. Bar graphs showing time spent (**E**), number of entries (**F**) in the open arms, (**G**) anxiety index, and representative trajectory diagrams (**H**) of Kmt2c^+/+^ and Kmt2c^+/−^ mice in the EPM test. The anxiety index in the EPM test was calculated as: (time spent in the open arms)/(total time spent in the open and closed arms) × 100%. (**A**–**C**) and (**E**–**G**): two-way ANOVA. Male Kmt2c^+/+^ mice: n = 12; Female Kmt2c^+/+^ mice: n = 12; Male Kmt2c^+/−^ mice: n = 13; Female Kmt2c^+/−^ mice: n = 13.

**Figure 6 biomolecules-15-01547-f006:**
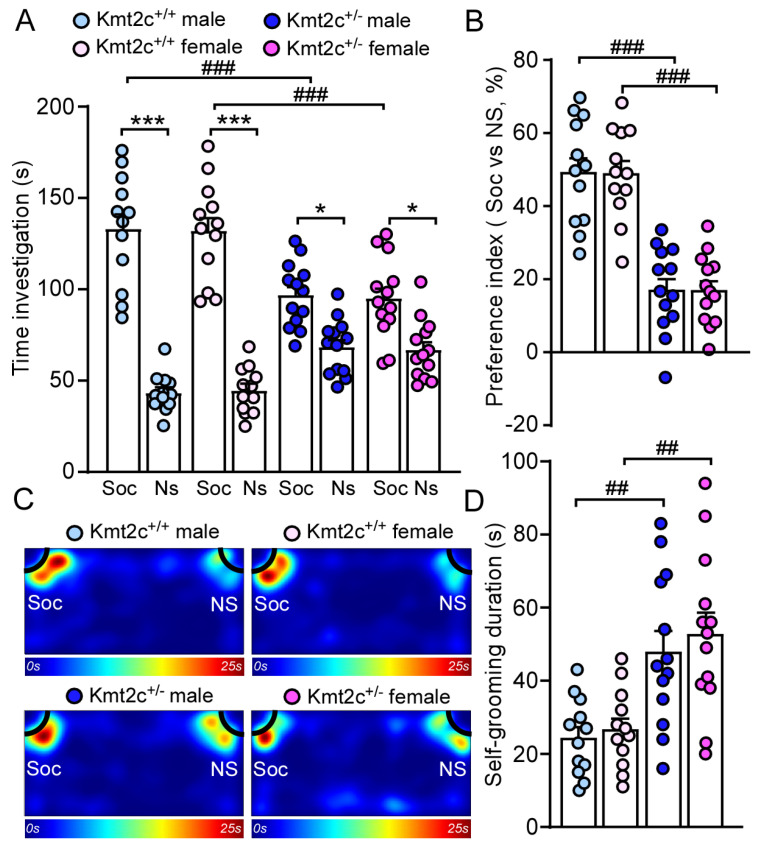
*Kmt2c* haploinsufficiency mice exhibit autism-like behavioral deficits. Bar graphs showing the time spent investigating social (Soc) or non-social (NS) stimulus (**A**) and social preference index (**B**) in the three-chamber sociability test of Kmt2c^+/+^ and Kmt2c^+/−^ mice. The preference index was calculated as: [time spent on social stimulus (Soc) − time spent on non-social stimulus (NS)]/[total time exploring the social and non-social stimuli (Soc + NS)] × 100%. (**C**) Representative heatmaps illustrating the time spent in different locations of the three chambers. Bar graphs showing the time spent self-grooming (**D**) in male and female Kmt2c^+/+^ and Kmt2c^+/−^ mice. * *p* < 0.05, *** *p* < 0.001, Soc versus NS; ^##^
*p* < 0.01, ^###^
*p* < 0.001, Kmt2c^+/−^ versus Kmt2c^+/+^. (**A**): three-way ANOVA; (**B**,**D**): two-way ANOVA. Male Kmt2c^+/+^ mice: n = 12; Female Kmt2c^+/+^ mice: n = 12; Male Kmt2c^+/−^ mice: n = 13; Female Kmt2c^+/−^ mice: n = 13.

**Figure 7 biomolecules-15-01547-f007:**
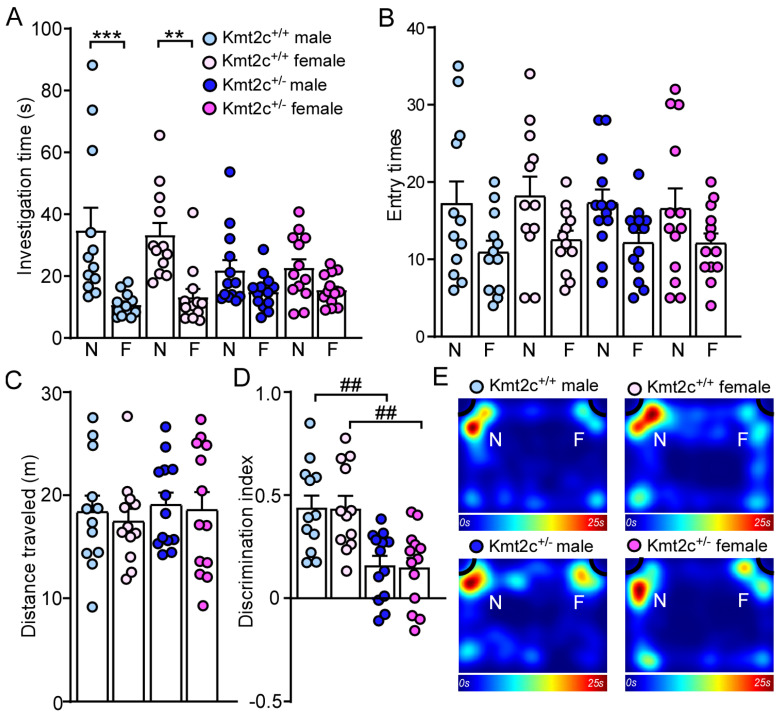
*Kmt2c* haploinsufficiency mice display novel object recognition deficits. Bar graphs showing the time spent (**A**) and entry times (**B**) to either novel object (N) or familiar object (F), total distance traveled (**C**), and discrimination index (**D**) in the NOR test of Kmt2c^+/+^ and Kmt2c^+/−^ mice. The discrimination index was calculated as: [time spent on novel object (N) − time spent on familiar object (F)]/[total time exploring both objects (N + F)] for the test session. (**E**) Representative heatmaps showing the time spent exploring the familiar and novel object during the NOR test. ** *p* < 0.01, *** *p* < 0.0001, novel object versus familiar object. ^##^
*p* < 0.01, Kmt2c^+/−^ versus Kmt2c^+/+^. (**A**,**B**): three-way ANOVA; (**C**,**D**): two-way ANOVA. Male Kmt2c^+/+^ mice: n = 12; Female Kmt2c^+/+^ mice: n = 12; Male Kmt2c^+/−^ mice: n = 13; Female Kmt2c^+/−^ mice: n = 13.

**Figure 8 biomolecules-15-01547-f008:**
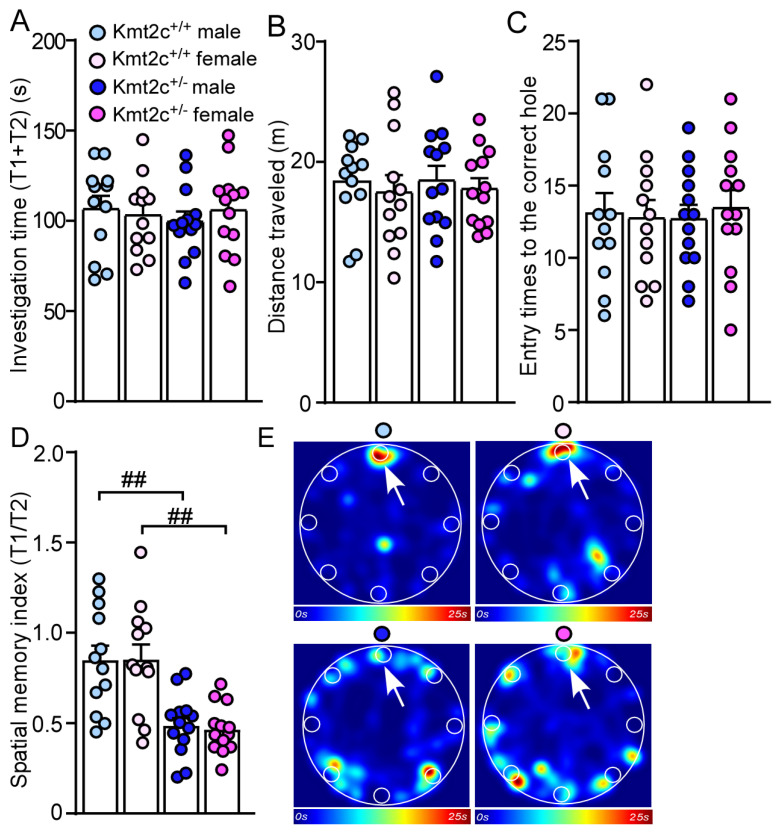
*Kmt2c* haploinsufficiency mice display spatial memory deficits. Bar graphs showing the total time spent investigating the correct hole and incorrect holes (**A**), distance traveled (**B**), entry time to the correct hole (**C**), and spatial memory index (T1/T2) (**D**) in the Barnes maze test of Kmt2c^+/+^ and Kmt2c^+/−^ mice. (**E**) Representative heatmaps illustrating the time spent in different locations of the arena during the memory phase (escape box removed). T1: time spent investigating the correct hole; T2: time spent investigating the other seven incorrect holes. The correct hole is pointed to by the arrow. ^##^
*p* < 0.01, Kmt2c^+/−^ versus Kmt2c^+/+^. (**A**–**D**): two-way ANOVA. Male Kmt2c^+/+^ mice: n = 12; Female Kmt2c^+/+^ mice: n = 12; Male Kmt2c^+/−^ mice: n = 13; Female Kmt2c^+/−^ mice: n = 13.

## Data Availability

The original contributions presented in this study are included in this article, further inquiries can be directed to the corresponding author.

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
