# Peer review of "Kmt2c/Mll3 Haploinsufficiency Causes Autism-like Behavioral Deficits in Mice"

_biomolecules, 2025, doi:10.3390/biom15111547_

Round 1

Reviewer 1 Report

Comments and Suggestions for Authors

In the manuscript entitled “Kmt2c/Mll3 haploinsufficiency causes autism-like behavioral deficits in mice" authors investigated the impact of germline Kmt2c haploinsufficiency on autism-like behavioral deficits in mice. They observed Kmt2c haploinsufficiency mice showed autism-like social deficits, increased self-grooming, and cognitive deficits, whereas no sign of anxiety and motor deficits. Overall, the KMT2C is important connecting link between several neurodevelopmental disorders, including autism spectrum disorder (ASD). However, the manuscript requires a major revision, including detailed methodology, improving sentences in the result section and in discussion. I think authors should include a section or few sentences in the discussion section of their findings of cognitive deficits in Kmt2c+/- mice.

Although, KMT2C related syndrome is ultra rare than ASD. However, Kmt2c haploinsufficiency is associated with acute myeloid leukemia, myelodysplastic syndrome, schizophrenia, ASD, Kleefstra syndrome 2, intellectual disability, hypotonia etc. The current findings suggest KMT2C gene maybe associated with some aspects of ASD, as authors find no motor deficits in the Kmt2c+/- mouse model. Therefore, I will refrain from saying Kmt2c+/- mouse as a model of ASD, but instead Kmt2c haploinsufficiency is associated with some behavioral impairment that is shared in several mental disorders including ASD, Angelman syndrome, and schizophrenia.

Line 14, KMT2C-related syndrome can be elaborate a bit.

Line 26, Sentence start with "Again" should be removed and write clearly which cognitive deficits have been observed.

Line 86-97, Behavioral test method is not sufficient. Several details are missing, such as how did they analyze the data, manual, or by using software? Which software etc.

Write the details of rotarod test. How many trails? And how is the final data achieved?

Duration of each test should also be mentioned.

Authors can check the mean speed of animals in OFT.

Lines 215-219, Figure 4, Be consistent with numbering of graphs, either write the numbering before text or after text.

Where is hindlimb clasping score data? No figures?

Figure 5C; Representative trajectory figures do not seem to be an open field arena, All the sides do not seem equal.

Is it trajectory of average mice from each group? Or only single mouse from each group?

If it's of single mouse, it should be removed or move to supplementary file.

Please show the normalized anxiety ratio/index. Like time/distance in center/total

Also, for EPM test, relative anxiety index should be analyzed. Example, Time in open arm/time in open arm + closed arm

Figure 6B, define how preference index has been calculated.

Line 305-318, include the figure numbers in text.

Figure 6, and 7; Make sure all the heat maps, representative trajectories are of all mice form each group, if not then either remove or move to the supplementary file.

A detail of social preference test required to put in the methodology, including duration of the test, familiarization.  Whether test mice were of same gender or opposite to experimental mice?

In the result section in text, all the results require adding proper figure numbers, including figure section (e.g.; Figure 5A, 5B).

Results sections need to be rewritten. Many results need proper elaboration. Some of the results are not significant, whereas others are significant, what does it indicate? For example, Kmt2c+/- mice showed normal motor function whereas total distance travelled were comparable (Lines 192-193).

Lines 235-237; what were the mild and high stress conditions?

Neither OF nor EPM test consider high stress conditions, unless authors included any other stressful conditions.

Lines 309-311; As shown in figure 7…..; But this is not shown in the figure 7, the sentence needs to be changed.

Line 312, How is discrimination index calculated? Stay consistent; either discrimination index or discrimination ratio both in text and figure.

Figure 7, insufficient data. Please provide other parameters of NOR test, such as time, distance, and number of entries.

Also, provide learning curve figure of the Barnes maze.

Define well number of training days, how many trials per day, and duration etc. in the method section.

Line 317, What is spatial memory index? What is T1/T2?

Also here provide other parameters of the probe day/memory day, such as time, distance, number of entries/crossings target area.

Reviewer 2 Report

Comments and Suggestions for Authors

dear colleagues i identified three issues in the paper. 

  • the major and most critical issues is that the study focuses on juvenile mice (6 wks) its equivalent to human adolescence but asd symptoms often emerge earlier around 2-3 years in human children. this is a major flaw in the study design however i think its still good start so discussing this as a limitation and suggesting longitudinal testing would address developmental trajectories and strengthen claims about neurodevelopmental roles. this need major discussion and implications. 
  • the link between kmt2c haploinsufficiency and asd-like behaviors is important findings for therapies. this need to be better discussed. the kmt2c gene is crucial for proper brain development. please check recent advancements in the field. 
  • the minor issue i would like to raise that all figures need to get enlarged. 

Round 2

Reviewer 1 Report

Comments and Suggestions for Authors

I am not convinced by the authors’ reply to some of my comments.

In Comment 2, I stated: “The current findings suggest that the KMT2C gene may be associated with certain aspects of ASD, as the authors report no motor deficits in the Kmt2c⁺/⁻ mouse model. Therefore, I would refrain from referring to the Kmt2c⁺/⁻ mouse as a model of ASD. Instead, I would suggest that Kmt2c haploinsufficiency is associated with behavioral impairments that are shared across several neurodevelopmental and psychiatric disorders, including ASD, Angelman syndrome, and schizophrenia.”

I still hold this view, as there is no evidence of hindlimb or forelimb deficits, nor any locomotor impairment, in the Kmt2c⁺/⁻ mice. Thus, referring to it as an ASD model is not sufficiently supported by the presented data.

My previous comments (5 and 6) were also not addressed sufficiently. I asked: “The behavioral test methods are not described in sufficient detail. Several important details are missing, such as how the data were analyzed—manually or using software—and, if software was used, which one.”

Regardless of whether the authors have described these methods in a previously published paper, it remains inconvenient and unclear to interpret the current results without inclusion of the analysis methodology. This critical information is missing from the present manuscript and should be provided to ensure reproducibility and transparency.

In my previous comment number 10, I asked “Figure 5C; Representative trajectory figures do not seem to be an open field arena; All the sides do not seem equal.”

All sides still do not seem to be equal.

In my previous comment, I requested that the authors present normalized anxiety ratios or indices (e.g., time or distance in center divided by total), and for the elevated plus maze (EPM) test, a relative anxiety index (e.g., time in open arms / [time in open arms + time in closed arms]).

The authors responded that “Kmt2c haploinsufficiency has no effect on anxiety-like behaviors; the normalized anxiety index will not provide more information than the current data in Figure 5.”

This response is not satisfactory. Instead of providing the requested normalized data, the authors argued that such analysis would not add further information. This reasoning is unconvincing and reduces confidence in the thoroughness of the behavioral analysis. For behavioral tests such as the EPM, presenting normalized data is preferable, as examining only a few raw parameters may not provide a complete or reliable interpretation of anxiety-related behavior.

For instance, in the authors’ own data (Figure 7), the parameter shown in Figure 7A (“investigation time”) reaches statistical significance, whereas Figures 7B and 7C (“entry times” and “distance traveling”) do not—despite all representing aspects of the same behavioral test. Similarly, in Figure 8, panel 8C shows no significant difference, whereas panel 8D shows significance within groups, although both parameters originate from the same behavioral paradigm. These inconsistencies further emphasize the importance of using normalized indices to achieve a more accurate and integrative understanding of behavioral outcomes.

Comment 12 was not addressed adequately. My original comment was: “In Figure 6B, please define how the preference index was calculated.”

Authors replied – “The calculation of preference index was included in our previous papers (ref. 24-26). The preference index was calculated, where time spent with one stimulus was subtracted from the time spent with the other stimulus and divided by the total time spent exploring both stimuli.”

However, this explanation remains unclear. Referring to “one stimulus” and “the other stimulus” does not specify which was used as the novel stimulus in the calculation. This distinction is essential for accurate interpretation of the behavioral results and should be explicitly stated in the methods or figure legend.

Author Response

Please see the attached response letter.

Reviewer 2 Report

Comments and Suggestions for Authors

no more comments. 

Author Response

Thanks a lot for the reviewer.